# Characteristics of Successful International Pharmacy Partnerships

**DOI:** 10.3390/pharmacy11010007

**Published:** 2023-01-01

**Authors:** Gina M. Prescott, Lauren Jonkman, Rustin D. Crutchley, Surajit Dey, Lisa T. Hong, Jodie Malhotra, See-Won Seo, Marina Kawaguchi-Suzuki, Hoai-An Truong, Elizabeth Unni, Kayo Tsuchihashi, Nubaira Forkan, Jeanine P. Abrons

**Affiliations:** 1Department of Pharmacy Practice, School of Pharmacy & Pharmaceutical Sciences, University at Buffalo, Buffalo, NY 14214, USA; 2Department of Pharmacy & Therapeutics, School of Pharmacy, University of Pittsburgh, Pittsburgh, PA 15261, USA; 3Department of Pharmacotherapy, Washington State University, Yakima, WA 98901, USA; 4College of Pharmacy, Roseman University of Health Sciences, Henderson, NV 89014, USA; 5Department of Pharmacy Practice, Loma Linda University School of Pharmacy, Loma Linda, CA 92350, USA; 6Department of Clinical Pharmacy, Skaggs School of Pharmacy & Pharmaceutical Sciences, University of Colorado, Aurora, CO 80045, USA; 7Department of Pharmacy Practice, Albany College of Pharmacy & Health Sciences, Albany, NY 12208, USA; 8School of Pharmacy, Pacific University Oregon, Hillsboro, OR 97123, USA; 9Department of Pharmacy Practice & Administration, University of Maryland Eastern Shore, Princess Anne, MD 21853, USA; 10Social, Behavioral, and Administrative Sciences, Touro College of Pharmacy, New York, NY 10027, USA; 11School of Pharmacy, University of Pittsburgh, Pittsburgh, PA 15261, USA; 12School of Pharmacy, University of Toronto, Toronto, ON M5S, Canada; 13Department of Pharmacy Practice & Science, College of Pharmacy, University of Iowa, Iowa City, IA 52242, USA

**Keywords:** global health, partnerships, pharmacy education, international, experiential education

## Abstract

Recommendations for global pharmacy collaborations are predominately derived from US institutions. This study utilized semi-structured interviews of global collaborators to assess important partnership components. Interviewees stated personal connections and understanding of each other’s programs/systems were key components. Additionally, collaborators indicate that mutual benefits between partners can exist without the requirement for bidirectional exchange of learning experiences, and request and value partners and learners who are culturally aware, global citizens. This structured interview approach provided key insight into how to develop mutually beneficial, sustainable partnerships and provides additional confirmation that the five pillars of global engagement align with an international audience.

## 1. Introduction

Global collaborations between colleges and schools of pharmacy have been increasing over the past decade [1,2]. Developing international collaborations can be time-consuming and challenging due to differences in policies, laws, infrastructure, and cultural considerations. These collaborations are often initiated by an institution in a high-income country and may or may not reflect a sustainable, mutually beneficial relationship. The American College of Clinical Pharmacy (ACCP) has developed an expert-based opinion paper on developing collaborations that are centered around five pillars including: sustainability, shared leadership, mutually beneficial partnerships, local needs-based care, and host-driven education [3]. There are also additional recommendations available for colleges and schools of pharmacy for planning and managing short-term medical mission trips and advanced pharmacy practice experiences [4,5,6,7]. These recommendations have been mainly derived from pharmacists/faculty in the United States (US). There are few recommendations based on direct input from non-US-based pharmacists. Furthermore, no available recommendations used a semi-structured interview approach. This study aims to provide standardized themes and direct feedback on collaborative agreements to US institutions from the perspectives of international partners.

## 2. Materials and Methods

### 2.1. Study Design

The American Association of Colleges of Pharmacy Global Education Special Interest Group (SIG) Global Outreach Committee (GOC) was comprised of eleven faculty members with and without global partnerships from US-based pharmacy institutions. After a review of the literature and several discussions, the committee developed an interview guide consisting of twenty-one questions organized into five categories, based on principles of developing international partnerships (See Table 1). These categories were further mapped to the five pillars of global health engagement. Committee members pre-tested the interview questions with an international collaborator and modified the guide for clarity and purpose. These modifications made the question categories more specific. This included replacing the category of “structure” with “student placement and student/faculty exchanges” and combining outcomes into the assessment category. An additional change included opening the interview with a broad question to set the tone for the interview. This question was “What factors might you consider when deciding whether to partner with a US University or College of pharmacy?” Finally, we rearranged our question order for better context based on the changes with the categories of questions. The research team reviewed the finalized interview guide to ensure coverage of all five pillars. The COREQ checklist was applied to the study (Appendix A) [8].

### 2.2. Participants

The GOC developed a list of pre-existing collaborators with a minimum of two to three collaborators from each WHO region (Africa, Americas, Eastern Mediterranean, Europe, South East Asia, and Western Pacific). Collaborators were known by the committee, either through direct affiliation or through professional pharmacy organization service and had experience in global collaborations. Collaborators were selected to be interviewed on a first-listed basis and upon response to a standardized recruitment email from the study investigators. Interviewers contacted collaborators at week one; if there was no response, one additional contact was made at week two, requesting an interview. Upon contact, collaborators were made aware of the study purpose and that investigators were members of the GOC. 

### 2.3. Data Collection

If the collaborator agreed to participate in the standardized interview, two investigators and the collaborator established a mutually agreed-upon time using Zoom Video Communications, Inc.™ (San Jose, CA, USA). The collaborator was also provided with an informed consent document to review prior to the interview. One investigator conducted the interview, while the other scribed. Both faculty members had prior experience with global collaborations and the faculty interviewers had previous experience conducting qualitative research. 

Upon initiating the interview, the interviewer obtained informed consent, changed the interviewee’s name to a number (for anonymity), and recorded the conversation for transcription purposes. Interviews lasted approximately one hour and were conducted between November 2021–June 2022. 

### 2.4. Data Analysis

Qualitative data analysis followed an inductive and deductive content analysis process. A sample size was not determined a priori, instead the GOC team aimed to include participants from diverse regions and continued interviews until thematic saturation was achieved meaning that no new concepts emerged from additional interviews. First, upon completion of each interview, an investigator reviewed and corrected the auto-transcriptions and removed any identifiable information before sharing the transcript with the full GOC. Next, a codebook was developed from the review of two de-identified transcripts using an open coding process. Using axial coding, the GOC then integrated the five pillars into the codebook. Two independent coders reviewed each transcript and discussed and reconciled any discrepancies. Additional codes were added to the codebook as required through the analysis process. The final thematic analysis was completed through an immersion-crystallization process whereby themes were coalesced from the codes and then interrogated by reviewing supportive quotes to ensure that themes aligned with the data. Overall, the analytic strategy proceeded through an iterative process requiring review of transcripts and team discussion where, the team developed and confirmed key themes. 

## 3. Results

Fourteen interviews, out of 17 attempts, were completed with 17 international partners representing 14 countries from five WHO regions: Africa (3), Americas (3), Europe (3), Eastern Mediterranean (2), and the Western Pacific (3).

Four themes were developed and connected to the pillars of global health engagement (Table 2).

The first theme highlighted how personal connections are critical to partnership development and sustainability. Participants gave examples of the importance of personal connections to developing international partnerships but noted that sustainability requires expansion beyond just one faculty member. Participants also described the value of partners with global health experiences as those partners have a better understanding of differences and similarities and adapt smoothly. Participants described building trust and rapport with the institution and individual partners as a crucial to relationship development. Participants noted the importance of a transition plan if the faculty lead at either institution leaves to ensure the partnership remains and thrives. Supportive quotes: 

“I know (the collaborator). There is a personal, level of trust, and we can get something done. I know MOUs are good to get institutional support. But, if we don’t have an inside person that we can deal with, the MOU becomes not useful.”

“When you sign an MOU, the partners need to agree to what would be the benefits in a partnership…I was lucky…those people that approached us, they knew upfront what they could contribute in terms of research, collaboration, a teaching collaboration, and so our biggest benefit was this.”

“I really look at relationships I have when I’m at different conferences, and we look to see whether or not there are any similarities within our institutions or where we can work together. And that’s where I start from.”

The second theme emphasized the importance of understanding each other’s programs and systems is essential to successful collaborative partnership. Participants noted that due to differences in pharmacy education around the globe, it is critical for partners to understand each other’s programs. This understanding includes recognition of: the timing of the academic year, holidays, and other cultural events or weather conditions; the evaluation strategies used for student assessment and expectations of students; the structure of the overall program including priorities and how students are trained; student levels (i.e., expectations for first year students versus third year students as well as different global educational programs such as Bachelor of Pharmacy versus Masters of Pharmacy versus Doctorate of Pharmacy); and prior knowledge before coming on site including priorities for the country, the university, and the profession. Further, it’s essential to consider the needs of partner sites, particularly in countries where resources are limited, including staff support to coordinate experiences and space/capacity restrictions. Finally, participants discussed the need to understand each other’s overall health systems, including how the training programs fit into the context of the overall health system. Adequate planning of logistics and coordination are critical to ensure the successful execution of local needs-based care and host-driven education.

“There are differences in practice, which I think being a preceptor, we need to help the students understand this and how to manage certain issues so that, at the end of the day, it will be a win-win situation.”

“Yeah, I think what is most complicated for us, more difficult for us to understand is how the US education system works. Like, you enter the university system, and then you take all these different classes and subjects and then you master or take your degree. I don’t know at some point you decide you want to, how do you say that, you want to major at some point. I don’t know”

“Institutional memory has been kept because the champions remained constant. Other people may think differently. And you remember also cultural exchange rates. You develop bonds, personal bonds, you know cultural bonds. Those are taken for granted, because you know, sometimes when there’s no funding the push to look for funding, this is coming from inside personally, because you have not developed your bond with the support group or with children by one part of the world, and so forth.”

The third theme was that mutual benefits can exist without bidirectional exchange, and those benefits may differ for each partner. Participants felt that equitable partnerships do not always require identical exchanges. Needs and opportunities vary between institutions. Participants discussed the benefits of layered learning with different levels of learners in other programs sharing what each knows. Participants discussed inequities in funding and resources between global sites and the impact on the physical exchanges of students. For instance, students from high-income countries may be able to support travel, while travel may be more difficult for students in low- or middle-income countries. Participants discussed sharing knowledge and best practices, including the experience of working with limited resources such as the WHO essential medicine lists and vice versa. Finally, participants noted that both parties might perceive and interpret successful collaboration differently. Moreover, it is vital to understand a particular institution’s priorities and needs to create a successful partnership and that mutual benefits sometimes carry different weights for each party. Supportive quotes:

“Partnerships definitely help strengthen our voice when it comes to health care within the country.”

“Joint publications. You and I published a paper together, or write a book chapter together. And then collaborate [with] students together. That is more than enough. And then, if we are released into the research, we can write a grant. Joint grant writing, and we can submit… Okay, otherwise we are jointly training students, so we might as well jointly develop curriculum.” 

“So there has to be trust instead of a document [i.e., MOU]…Once you have a good partner, that is willing to work on things.”

“I had two students from (institution), they helped me put together the pharmacy program here. They used their knowledge of pharmacy and my knowledge, and then the government pharmacist knowledge… If it wasn’t for that, there would be a massive shortage of pharmacists… right now.”

Finally, partners identified open mindedness, adaptability, global citizenship, and cultural and structural awareness as essential qualities for partners and learners. Partners need to be thoughtful about preparing learners who are open-minded, can critically evaluate assumptions, can identify collectivist versus individualist perspectives, and maintain a global rather than ethnocentric mindset. These qualities are necessary for effective partnerships and high-quality experiences for learners.

“I wish that going forward when we have a critical mass of mentors that understand global health, what international exchanges is all about. We will do all the technical, but we must know that we are dealing with a holistic situation…and a cultural exchange aspect.”

“I think our biggest expectation is that they should be willing to learn new things because …working in a low to middle income country you might have all the knowledge, you might know the best way to treat a patient, but that medicine, all that treatment is not available in the country so how do you juggle that. So in that instance, that you must be willing to listen, learn how we do things, and it’s not necessarily that it’s the best way to do things but it’s the only way we have and we have to make that work. We discussed this earlier, you know, is to be culturally sensitive because we dealing also with this issue of colonialism. And you know as an academic bringing in US students I get confronted at management level about colonialism…They need to be sensitive towards that and not get offended……It’s trying to give them that opportunity to give a plan of how they would treat this patient in the US, but then to bring the context to them the challenges that we face with access to medicine and then develop a new plan and that willingness to learn and change becomes important.”

## 4. Discussion

This is the first study we are aware of that provides a unique insight into pharmacy partnerships from internationally based key informants representing different regions from around the world. Interview approaches with international partners have been previously utilized to assist with important pharmacist advancements, including pharmacists as immunizers and understanding the role of continuous professional development in the health professions [9,10]. Similarly, US schools of pharmacy utilized a network approach with healthcare systems to determine how to best meet the needs of their partners and maximize commitments to each other [11]. Finally, the WHO commissioned a report that explored themes of global interprofessional collaboration in different WHO regions to determine how collaborative practice was defined across the world [12]. A strength of our study was the provision of an open dialogue for global collaborators to discuss their viewpoints on essential components of global partnerships. While the interview had structured questions, interviewees could speak freely on any aspect to partnership development that they felt was important. The inductive thematic analysis of 17 interviews revealed many key issues that were able to be linked to the five pillars of global health engagement [3].

Personal connections and trust between partners were considered key factors in contributing to the sustainability of programs. This is in accordance with the ACCP pillars on ethical engagement for sustainability. The reputation of an individual and their ability to navigate cross-cultural communication as well as understand the collaborators’ perspective of mutual benefit may be as important as the reputation of a particular institution. Global partnerships are often started through individual connections, and while the literature on this is mixed with some recommending utilizing pre-existing relationships already established at your college or university, the key factor here is likely trust with the intent to develop a sustainable partnership [6,13].

Personal connections were again mentioned along with a shared understanding of each other’s systems and programs. These connected to the ACCP pillar of shared leadership. Developing a clear goal for each institution based on their local healthcare or academic setting was important to the interviewees. This is consistent with literature on the focusing on needs-based care [6,13]. Importantly, participants discussed challenges that they have faced with students’ work being evaluated using US-based evaluations. Interviewers discussed the need for cultural and structural sensitivity to avoid using a high-income country lens to look at local needs-based interventions rather than looking to partners for their feedback on the work of students in non-US settings. Examples of long-term, sustainable global partnerships focused on local needs-based care, have demonstrated improvements in patient outcomes and enhancing workforce training [14,15,16,17,18]. In Zimbabwe, a focus on HIV clinical pharmacology and post-doctoral training programs has improved research training, optimization of antiretroviral use, and development of national treatment guidelines [14,15]. In Kenya, improvements through inpatient and outpatient pharmacy settings have improved clinical pharmacy training and workforce development [16]. In West Africa, an antimicrobial stewardship train-the -trainer program improved workforce capacity in this identified area of need [17]. Finally, in Thailand, improvements to advance clinical pharmacy education led to pharmacists’ recognition in national initiatives/practice guidelines, post-graduate programs, and pharmacy workforce [18].

The third theme centered around the concept that mutually beneficial goals did not necessarily mean that an identical experience was needed in exchange of educational experiences. This was connected to the mutually beneficial partnership and local needs-based goals stemming from the consideration that partners discussed what their institutions valued and needed for their learners. While our research team initially thought it should be equitable (as in physically exchanging on a student for student basis) our partners reminded our team that equity meant more than just student exchange. Collaboration outcomes such as publications, the establishment of programs, and positive impacts to a country’s health system represented alternative markers of collaboration success. All the sustainable programs previously mentioned have cited these markers as well in their measures of successful partnerships [14,15,16,17,18]. Travel was not an assumed component of the collaborations, and in some instances, partners felt that student travel should be deprioritized as it has a limited impact on the institution. Bringing US students to areas in which clinical practice by pharmacists is growing was also discussed as a beneficial opportunity for collaborative development and has been discussed in the literature as well [6]. An important component to the mutually beneficial partnership pillar includes providing transparency, developing shared goals, and recognizing a level of equity and mutual respect [3]. In Taiwan, development of clinical pharmacist services and a residency program through collaborations has been recognized as an output of successful partnerships. This can help to improve the overall value of pharmacist globally and with patient care [19,20].

The fourth theme discussed the need for partners and learners to be open-minded, adaptable, and in general, global citizens. This was a common theme amongst all interviewees. The concept of developing students who are global citizens is a newer phenomenon in academia broadly and within pharmacy school curricula specifically [21,22]. The growing acknowledgement of global interconnectedness reinforced through recent events, including the pandemic has accelerated a priority on decolonization and identifying and addressing ethnocentrism [22,23]. The Consortium of Universities for Global Health (CUGH) has competencies related to the development of a global health citizen and created a toolkit to assist with competency attainment [24,25]. Tools, such as the Cultural Intelligence Tool (CQ), the Intercultural Development Inventory (IDI), or others could be explored to help further create prospective mindfulness, and reflective abilities of learners before exchanges [26,27,28]. In preparing to engage in shared work and learning, elements that should be presented to learners and discussed to ensure common understanding between collaborators include cultural humility, cultural nuances or differences, and structural factors that may influence the collaboration, health system, and educational experiences/dynamics.

One area that was not mentioned as frequently but was integrated into many recommendations from participants was the role of an interprofessional approach to global exchanges. This is likely due to a few reasons. The importance of interprofessional education and collaborative practice for the global healthcare workforce has been recognized by the WHO, and the pharmacist’s role in an interprofessional environment is also noted in the 2022 International Pharmaceutical Federation Global Competency Framework for Educators & Trainers in Pharmacy [22,29]. So, while there is support for these initiatives and improvements have been seen in perspective towards interprofessional education and practice, these experiences still vary widely across the world [30,31]. The lack of specific commentary in our study may be due to the interview questions not specifically addressing interprofessional approaches or that it was implied by discussing advanced practice models.

Limitations to our research include the small number of interviewees per geographic location, although we continued interviews until thematic saturation was achieved. In addition, interviewed individuals included existing partners, thereby the study has the potential for social desirability bias.

Future areas of research include providing a series of perspectives on individual WHO regions, expansion on the topic of interprofessional educational efforts and how successful collaborations achieve these goals, and how didactic education can be linked to experiential education with both the host and partner institutions in various WHO geographical regions.

## 5. Conclusions

Overall, global perspectives on successful and sustainable partnerships are consistent with expert-based guidance and descriptions published from sustainable programs. While most partners agreed that mutual respect and understanding of each other’s programs and goals are essential, how a successful partnership is achieved and measured can vary. Following a local needs-based approach may assist with developing partnerships.

## Figures and Tables

**Table 1 pharmacy-11-00007-t001:** Interview Guide.

Category	Questions	Pillar ^1^
General	What factors might you consider when deciding whether to partner with a US University/College of Pharmacy?	1–5
Process	What challenges might exist in creating a formal agreement, such as a memorandum of understanding (MOU)?	2–3
	What would you expect to see in such an agreement?	1–5
Student Placement	How do you weigh the benefits or harms of student exchanges or placements at your institution?	1–5
	What preparation should students be aware of when traveling to a host country?	
	Can you tell me about factors that impact the number/type/time of student placements?	
	What factors are consistent from year to year and what factors are dependent upon the current context?	
	What factors would influence the time of year that would be appropriate for students to travel?	
	What supervision would be expected for students participating in international placements? From the host site? From the institutional site sending students?	
	In what types of experiences would you consider involving students at your institution or partner institutions?	
	What is expected of the US students while at the site and what contributions could they make?	
	From your perspective, what hands-on roles do you feel are most appropriate for US-trained students	
	What strategies have you seen that might allow for student involvement?	
	What limitations or challenges exist that would limit international learners at your site?	
Student or Faculty/Staff Exchange	In thinking about mutually beneficial partnerships, what would be the value for your students or faculty/staff to receive?	3
	What might be some challenges that exist in developing bilateral exchanges for either students or faculty/staff?	1–5
Assessment	How do you evaluate exchange students on rotation?	2, 5
	What are your thoughts about completing evaluations for US students?	3, 5
	What would make you more comfortable or less comfortable with evaluating US students?	3, 5
	How would you define a successful collaboration between a US school of pharmacy and your institution?	1–5
	What measures define a valuable experience for students? For the site? For others?	1–5

^1^ Pillars include: 1—Sustainability, 2—Shared Leadership, 3—Mutually Beneficial Partnerships, 4—Local Needs Based Care, 5- Host Driven Education.

**Table 2 pharmacy-11-00007-t002:** Global Health Interviews Major Themes and Supportive Pillars.

Theme	Global Health Pillar
Personal connections are critical to partnership development and sustainability	Sustainability, Shared Leadership
Understanding of each other’s programs and systems is essential for a successful collaborative partnership	Shared Leadership, Host-driven Education
Mutual benefits can exist without bidirectional exchange and may be different for each partner	Mutually Beneficial Partnerships, Local Needs-Based Care, Host-Driven Education
Key qualities for supporting overall collaboration and partnership include open-mindedness, adaptability, global citizenship, and cultural/structural awareness	Sustainability, Mutually Beneficial Partnerships, Host-Driven Education, Shared Leadership, and Local Needs-Based Care

## Data Availability

Not applicable.

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
