# Peer review of "Characteristics of Successful International Pharmacy Partnerships"

_pharmacy, 2023, doi:10.3390/pharmacy11010007_

Round 1

Reviewer 1 Report

Line 54 - 55. Section 2 (Materials and Methods)

“After a review of 53 the literature and several discussions, the committee developed an interview guide consisting of twenty-one questions organized into five categories. based on principles of developing international partnerships (See Table 1).” The word “based” should be capitalized since it comes after a full stop.

Line 57 – 58. “Committee members piloted the interview questions with an international collaborator and modified the guide for clarity “

I believe that the correct term to use to describe this process should be “pretest’ instead of ‘pilot test.’ These processes are completely different.

The two terms are often used interchangeably, but there is one critical difference between the two. In a pretest, you only test one or a few components of the research study on a small fraction of your intended sample size. Pretest would require a few or one individual. During a pilot, you conduct the research study entirely but on a smaller sample size. The fact that only one individual reviewed the survey would indicate that it was a pretest and not a pilot test.

I invite the authors to use the appropriate terminology.

 Line 57 – 60 “ Committee members piloted the interview  questions with an international collaborator and modified the guide for clarity and purpose. The research team reviewed the finalized interview guide to ensure coverage of all  five pillars.”

The authors need to specify the changes made after the international collaborator reviewed the questionnaire. This will help the readers better assess how much the questionnaire was improved from its original version.

 Line 83 – 86

“The GOC then completed a thematic analysis by reviewing the codes and supportive quotes. Through an iterative process requiring review of transcripts and team discussion, the team developed and confirmed key themes.”

 Thematic analysis is a widely used, yet often misunderstood, qualitative data analysis method. The authors may need to provide additional information about this process. Specifically, the authors need to illustrate each step of the thematic analysis process. From my perspective, it will prove difficult to replicate this work, given the preliminary nature of the provided information.

Line 64 – 67

‘The GOC developed a list of pre-existing collaborators with a minimum of two to 64 three collaborators from each WHO region (Africa, Americas, Eastern Mediterranean, Eu-65 rope, South-East Asia, and Western Pacific). Collaborators were selected to be interviewed 66 on a first-listed basis and upon response to a standardized recruitment email from the 67 study investigators.’

The authors may strengthen this section by discussing the sample size. There is a need to specify whether the sample size was calculated or determined for this work, considering that the authors had a-priori information about the target group of respondents.

Author Response

Point 1: Line 54 - 55. Section 2 (Materials and Methods)

“After a review of 53 the literature and several discussions, the committee developed an interview guide consisting of twenty-one questions organized into five categories. based on principles of developing international partnerships (See Table 1).” The word “based” should be capitalized since it comes after a full stop.

Response 1: The word “based” was kept, but the period was changed to a comma for better sentence structure.

Point 2: “Committee members piloted the interview questions with an international collaborator and modified the guide for clarity “I believe that the correct term to use to describe this process should be “pretest’ instead of ‘pilot test.’ These processes are completely different.

The two terms are often used interchangeably, but there is one critical difference between the two. In a pretest, you only test one or a few components of the research study on a small fraction of your intended sample size. Pretest would require a few or one individual. During a pilot, you conduct the research study entirely but on a smaller sample size. The fact that only one individual reviewed the survey would indicate that it was a pretest and not a pilot test.

I invite the authors to use the appropriate terminology.

Response 2: The term piloted was changed as recommended to “pretest”

Point 3:  Line 57 – 60 “Committee members piloted the interview questions with an international collaborator and modified the guide for clarity and purpose. The research team reviewed the finalized interview guide to ensure coverage of all five pillars.”

The authors need to specify the changes made after the international collaborator reviewed the questionnaire. This will help the readers better assess how much the questionnaire was improved from its original version.

Response 3: (Lines 60-67): Added to the paper, “These modifications made the question categories more specific. This included replacing the category of “structure” with “student placement and student/faculty exchanges” and combining outcomes into the assessment category. An additional change included opening the interview with a broad question to set the tone of the interview. This question was “What factors might you consider when deciding whether to partner with a University or College of pharmacy in the US?” Finally, we rearranged our question order for better context based on the changes with the categories of questions. 

Point 4: Line 83 – 86 “The GOC then completed a thematic analysis by reviewing the codes and supportive quotes. Through an iterative process requiring review of transcripts and team discussion, the team developed and confirmed key themes.”

 Thematic analysis is a widely used, yet often misunderstood, qualitative data analysis method. The authors may need to provide additional information about this process. Specifically, the authors need to illustrate each step of the thematic analysis process. From my perspective, it will prove difficult to replicate this work, given the preliminary nature of the provided information.

 Response 4: We have addition the following items (Page 4, Lines 106-113): ” Next, a codebook was developed from the review of two de-identified transcripts using an open coding process. Using axial coding, the GOC then integrated the five pillars into the codebook. Two independent coders reviewed each transcript and discussed and reconciled any discrepancies. Additional codes were added to the codebook as required through the analysis process. The final thematic analysis was completed through an immersion-crystallization process whereby themes were coalesced from the codes and then interrogated by reviewing supportive quotes to ensure that themes aligned with the data. Overall, the analytic strategy proceeded through an iterative process requiring review of transcripts and team discussion where, the team developed and confirmed key themes.” In addition, we added component in for the COREQ checklist in Appendix A, per another reviewer’s comments.

Point 5: Line 64 – 67 ‘The GOC developed a list of pre-existing collaborators with a minimum of two to 64 three collaborators from each WHO region (Africa, Americas, Eastern Mediterranean, Eu-65 rope, South-East Asia, and Western Pacific). Collaborators were selected to be interviewed 66 on a first-listed basis and upon response to a standardized recruitment email from the 67 study investigators.’

The authors may strengthen this section by discussing the sample size. There is a need to specify whether the sample size was calculated or determined for this work, considering that the authors had a-priori information about the target group of respondents.

Response 5: We have adjusted the manuscript and included the following response (Page 4, Line 100-103): “Qualitative data analysis followed an inductive and deductive content analysis process. A sample size was not determined a priori, instead the GOC team aimed to include participants from diverse regions and continued interviews until thematic saturation was achieved meaning that no new concepts emerged from additional interviews.” In addition, we added component in for the COREQ checklist in Appendix A, per another reviewer’s comments.

Reviewer 2 Report

Row 55: ”categories. based” –  ”categories, based”

Materials and methods: the time period of the data collection is missing

Results: the sociodemographic and professional characteristics of the participants should be included

Table 2: ” differences in in practice” – differences in practice”

Table 2: listing the answers almost per se is not an appropriate way to present the results in a tabulated format, a scientific interpretation of the answers is needed: finding keywords, following the guidelines of the Consolidated criteria for Reporting Qualitative research (COREQ), etc.

Comparison of the results with other similar studies is necessary at the conclusion section (data about other networks with similar profile, etc.)

Row 142-144: it is true, but studies regarding similar research areas still are available:

https://pubmed.ncbi.nlm.nih.gov/31213308/

https://pubmed.ncbi.nlm.nih.gov/31167345/

Author Response

Point 1: Row 55: ”categories. based” –  ”categories, based”

Response 1: The period has been changed to a comma

Point 2: Materials and methods: the time period of the data collection is missing

Response 2: The time period has been added: “Interviews lasted approximately one hour and were conducted between November 2021- June 2022.”(Page 4, Line 94-95)

Point 3:  Results: the sociodemographic and professional characteristics of the participants should be included

Response 3: While we recognize that this may be important, we have not provided all the sociodemographic information due to our small sample size, as we wish to make sure our sources stay de-identified. We did add in that all had experience in global collaborations and added information to the COREQ checklist per your future recommendation (Page 3, Line 75-76; Appendix A).

Point 4: Table 2: ” differences in in practice” – differences in practice”

Response 4: This has been fixed.

Point 5: Table 2: listing the answers almost per se is not an appropriate way to present the results in a tabulated format, a scientific interpretation of the answers is needed: finding keywords, following the guidelines of the Consolidated criteria for Reporting Qualitative research (COREQ), etc.

Response 5: We have adjusted the Table. Additionally, we have included the COREQ checklist as Appendix A. In addition, those items not originally in the manuscript, that are in the checklist have now been included in the manuscript. Thank you.

Point 6: Comparison of the results with other similar studies is necessary at the conclusion section (data about other networks with similar profile, etc.

Response 6: Thank you. I have updated the discussion and the conclusion based on the studies below and have added additional references as well throughout these areas.

Point 7: Row 142-144: it is true, but studies regarding similar research areas still are available:

https://pubmed.ncbi.nlm.nih.gov/31213308/

https://pubmed.ncbi.nlm.nih.gov/31167345/

Response 7: Thank you. I’ve adjusted the statement to include the studies that you have listed.
